# Estimation of Minced Pork Microbiological Spoilage through Fourier Transform Infrared and Visible Spectroscopy and Multispectral Vision Technology

**DOI:** 10.3390/foods8070238

**Published:** 2019-07-01

**Authors:** Lemonia-Christina Fengou, Evgenia Spyrelli, Alexandra Lianou, Panagiotis Tsakanikas, Efstathios Z. Panagou, George-John E. Nychas

**Affiliations:** Laboratory of Microbiology and Biotechnology of Foods, Department of Food Science and Human Nutrition, Agricultural University of Athens, Iera Odos 75, 11855 Athens, Greece

**Keywords:** spectroscopy, non-invasive sensors, microbiological spoilage, minced pork

## Abstract

Spectroscopic and imaging methods coupled with multivariate data analysis have been increasingly studied for the assessment of food quality. The objective of this work was the estimation of microbiological quality of minced pork using non-invasive spectroscopy-based sensors. For this purpose, minced pork patties were stored aerobically at different isothermal (4, 8, and 12 °C) and dynamic temperature conditions, and at regular time intervals duplicate samples were subjected to (i) microbiological analyses, (ii) Fourier transform infrared (FTIR) and visible (VIS) spectroscopy measurements, and (iii) multispectral image (MSI) acquisition. Partial-least squares regression models were trained and externally validated using the microbiological/spectral data collected at the isothermal and dynamic temperature storage conditions, respectively. The root mean squared error (RMSE, log CFU/g) for the prediction of the test (external validation) dataset for the FTIR, MSI, and VIS models was 0.915, 1.173, and 1.034, respectively, while the corresponding values of the coefficient of determination (R^2^) were 0.834, 0.727, and 0.788. Overall, all three tested sensors exhibited a considerable potential for the prediction of the microbiological quality of minced pork.

## 1. Introduction

The current safety and quality controls in the food chain are largely relying on chemical and microbiological analyses, which are applied in the context of sampling regimes delineated by regulatory requirements. This is the case with meat and meat products where the microbiological analyses involve either conventional microbiology approaches (e.g., colony counting methods) or molecular techniques that are considered reliable and accurate [1,2]. However, these analyses are time-consuming, laborious and provide retrospective results. Thus, the development of novel analytical technologies for the estimation of the microbial load in food would be very important for the rapid screening of a high number of food samples in real-time, especially for food industries and retailers.

The last 15 years, UV, Visible, Infrared, Raman, and hyper-/multi-spectral imaging methods have been used thoroughly to investigate possible relationships between spectra and microbial populations. The ultimate goal of such research efforts is the development of methods that will allow for the rapid and non-invasive assessment of the microbiological quality and safety of foods, with minimum or even no sample preparation [3]. However, the enormous amount of acquired data derived from spectroscopic measurements is far more complex than the datasets generated in the context of traditional microbiological analyses, and, as such, usually requires processing before the results can be interpreted. Indeed, with the evolution of data science and machine learning approaches, novel computational methods emerged to rapidly provide information related to food safety and/or quality (including categorization of foods with regard to spoilage), through the development of classification or regression models using spectral or imaging data for model training and validation [4]. Specifically, machine learning approaches have been used for the qualitative or quantitative evaluation of microbial levels in meat and meat products, fish, fruits, dairy products, and vegetables [5,6,7,8,9,10,11,12].

The aim of this work was the estimation of the microbial population of minced pork using Fourier transform infrared (FTIR) and visible (VIS) spectroscopy as well as multispectral image (MSI) acquisition, through the implementation of multivariate data analysis. In this framework, models were trained (developed) and validated (tested) individually for each one of the sensors under study (i.e., FTIR, VIS and MSI), whereas a “naïve” data fusion scheme also was investigated.

## 2. Materials and Methods

### 2.1. Minced Pork Samples Preparation and Storage Conditions

Minced pork was purchased from a local butchery in Athens, Greece (where it was ground) and transported to the laboratory with minimal delay, i.e., within 30 min. Minced pork was stored at 0 °C until the packaging process was completed. Packaging took place under a laminar flow cabinet, in order for aseptic conditions to be ensured. Meat portions of ca. 100 g were weighed, patties were prepared and packaged in styrofoam trays: Two patties were placed in each styrofoam tray and were wrapped with cling film as previously described by Panagou et al. [13]. The minced pork patties were stored at different isothermal conditions (4, 8, and 12 °C) and under dynamic temperature conditions (i.e., periodic temperature changes from 4 to 12 °C) in high-precision (± 0.5 °C) programmable incubators (MIR-153, Sanyo Electric Co., Osaka, Japan) for a maximum time period of 14 days (during storage at 4 °C). Two independent minced pork batches were purchased at distinct times and used in the storage experiments, corresponding to two different experimental replicates. Minced pork samples were analyzed at the time of arrival to the laboratory (time-zero) and at regular time intervals during storage, depending on the applied storage temperature; specifically, analyses were carried out at 10 or 14 h intervals until 110 h of storage (sampling times: 0, 14, 24, 38, 48, 62, 72, 86, 96, and 110 h), and at 24 h interval thereafter. The experiments were concluded (i.e., final time points of analyses) at 158, 230, 302, and 230 h of storage at 4 °C, 8 °C, 12 °C and dynamic temperature conditions, respectively. On each sampling interval, duplicate patties (i.e., biological replicates) contained in a randomly selected tray were subjected to: (i) microbiological analyses and pH measurement, (ii) FTIR spectroscopy measurements, (iii) VIS spectroscopy measurements, and (iv) MSI acquisition. In total, 228 different minced patties were used in the aforementioned analyses.

### 2.2. Microbiological Analyses and pH Measurement

For the purpose of microbiological analyses, 25 g portions of minced pork patties were added aseptically to 225 mL of sterile quarter strength Ringer’s solution (Lab M Limited, Lancashire, United Kingdom) in 400 mL sterile stomacher bags (Seward Medical, London, United Kingdom) and homogenized in a Stomacher apparatus (Lab Blender 400, Seward Medical) for 60 s at room temperature. Appropriate serial decimal dilutions (0.1 mL aliquots) were prepared and surface plated on the following agar media: (i) Tryptic glucose yeast agar (Plate Count Agar; Biolife, Milan, Italy) for the enumeration of total mesophiles (total viable counts, TVC), after incubation of plates at 25 °C for 72 h [14]; (ii) Pseudomonas Agar Base with selective supplement cephalothin-fucidin-cetrimide (Lab M Limited) for the enumeration of presumptive *Pseudomonas* spp., after incubation of plates at 25 °C for 48 h; (iii) Streptomycin Thallous Acetate-Actidione Agar (STAA, Biolife) for the enumeration of *Brochothrix thermosphacta*, after incubation of plates at 25 °C for 48 h; and (iv) Rose Bengal Chloramphenicol Agar (RBC, Lab M Limited) for the enumeration of moulds and yeasts, after incubation of plates at 25 °C for 5 days. Furthermore, 1 mL aliquots of appropriate serial decimal dilutions of the minced pork patties’ homogenates also were pour plated in the following agar media: (i) de Man, Rogosa and Sharpe (MRS) agar (Biolife) for the enumeration of presumptive lactic acid bacteria (LAB), after incubation of plates at 30 °C for 72 h; and (ii) Violet Red Bile Glucose (VRBG) agar (Biolife) for the enumeration of bacteria of the family Enterobacteriaceae, after incubation of plates at 37 °C for 24 h.

After enumeration of microbial colonies, the microbiological data were converted to log colony forming units per gram of meat, i.e., log CFU/g.

Upon completion of the microbiological analyses, the pH values of the meat samples’ homogenates also were measured using a digital pH meter (RL150, Russell pH, Cork, Ireland) with a glass electrode (Metrohm AG, Herisau, Switzerland).

### 2.3. Sensors

#### 2.3.1. FTIR Spectroscopy

FTIR spectral data were collected using a ZnSe 45° HATR (Horizontal Attenuated Total Reflectance) crystal (PIKE Technologies, Madison, Wisconsin, United States), and a FTIR-6200 JASCO spectrometer (Jasco Corp., Tokyo, Japan). The crystal used has a refractive index of 2.4 and a depth of penetration of 2.0 μm at 1000 cm^−1^. Spectra were collected over the wavenumber range of 4000 to 400 cm^−1^ using the Spectra Manager™ Code of Federal Regulations (CFR) software version 2 (Jasco Corp.), by accumulating 100 scans with a resolution of 4 cm^−1^ and a total integration time of 2 min.

A small portion from each meat sample was transferred to the crystal plate, covered with a small piece of aluminum foil, and then pressed to ensure the best possible contact with the crystal. Prior to the measurements of the samples, reference spectra were acquired using the cleaned blank (no added sample) crystal. After each measurement, the crystal’s surface was cleaned, first with detergent and distilled water and then with analytical grade acetone, and dried using lint-free tissue. As previously practiced [15], the FTIR spectra that were ultimately used in further analyses were in the approximate wavenumber range of 1800 to 900 cm^−1^.

#### 2.3.2. VIS Spectroscopy

The VIS spectroscopy measurements were conducted using the Hamamatsu C12880MA (Hamamatsu Photonics K.K., Shizuoka, Japan) spectrometer, which has a spectral range from 340 to 850 nm, high sensitivity and spectral resolution of 15 nm. Meat samples were placed in Petri dishes and 10 different spectral measurements (absorbance values) were acquired at different spots of the samples, and their average value was used in order for the inherent absorbance diversity to be taken into account. A more detailed description of the applied procedure can be found in Tsakanikas et al. [11].

#### 2.3.3. Image Acquisition and Pre-Processing

Multispectral images were acquired using the VideometerLab system [16]. This instrument acquires multispectral images in 18 different, non-uniformly distributed wavelengths ranging from UV (405 nm) to short wave NIR (970 nm). The procedure for image acquisition and segmentation has been described in detail previously [13,17]. Briefly, prior to image acquisition, the system was subjected to a light set up procedure known as “autolight” and calibrated radiometrically and geometrically. The minced meat samples were transferred to Petri dishes, placed inside the Ulbricht sphere and multispectral images of the samples’ surface were taken. In order for redundant information (related to the background, fat, and not to the sample) to be removed, image segmentation was performed using the VideometerLab system software (version 2.12.39). The contrast between the sample material and the other irrelevant objects was maximized, in order to enable a threshold operation. Then, canonical discriminant analysis was employed, resulting in the production of a segmented image. After the segmentation process of each image, the mean reflectance spectra within the informative area (i.e., mean intensity of pixels) along with the corresponding standard deviation values were calculated.

### 2.4. Data Analysis

#### 2.4.1. Microbiological Data Analysis

Aiming at the more detailed characterization of the microbial growth kinetic behavior, the primary model of Baranyi and Roberts [18] was fitted to the microbiological data corresponding to the populations of the microorganisms contributing considerably to minced pork spoilage. Specifically, the growth kinetic parameters that were estimated for selected microbial groups included the lag time (λ), the maximum specific growth rate (μ_max_), the initial microbial population (y_0_) and the maximum population density (y_end_).

#### 2.4.2. Spectral Data Analysis and Correlation with Microbiological Data

The objective of this analysis was to investigate the potential of the aforementioned sensors’ spectral data to provide accurate estimates of the microbial load of minced pork samples. The acquired data were first pre-processed in order to enhance their quality and combat with the correlated information across the different wavelengths and their inherent, due to the acquisition process, multiplicative noise. Then, partial least squares regression (PLSR) models were developed for each sensor and evaluated. The PLSR was the method of choice since it has been used widely in the food research field and is thus, considered as the standard technique for spectroscopic analysis. Moreover, it is suitable for datasets where the number of variables is greater than number of samples, and also when collinearity among variables is apparent and where variable selection is then proposed [19,20].

Specifically, the acquired FTIR spectra were pre-processed with the Savitzky–Golay smoothing numerical algorithm [21] with a second-order polynomial and a 9-point window. The VIS and the MSI spectral data were pre-processed using Standard Normal Variate (SNV), which intends to correct spectra for light scatter and adjust for baseline shifts among samples [22]. Spectral/imaging data (FTIR, VIS, and MSI) were distinctively used as input variables (X) and TVC as output (prediction) variables (Y). Datasets were split into training set, for developing the model (n = 170 samples), and test set, for external validation of the model (n = 58). Specifically, the PLSR models were trained with the data generated during storage of minced pork patties at isothermal conditions, and were validated against the data corresponding to storage at dynamic temperature conditions, so as to simulate storage/distribution temperature conditions encountered in real life. For the training process, leave one out cross validation was employed for the selection of significant latent variables and for avoiding models’ overfitting to the data. Data sets derived from FTIR and VIS spectroscopy were subjected to Martens Uncertainty test as described in Westad and Martens [23], in order to simplify the final model and make it more reliable. At this point, it should be noted that an attempt to combine all spectra (data concatenation) and build an elementary data fusion scheme also was made. Briefly, the spectra acquired from each sensor were baseline corrected, normalized into the range [0, 1] and were concatenated to set up a single array, representing each sample. Then, a PLSR model was developed, as described earlier, where the data were pre-processed using the SNV approach. Data analysis was performed using the chemometric software The Unscrambler^®^ ver. 9.7 (CAMO Software AS, Oslo, Norway).

## 3. Results and Discussion

### 3.1. Microbiological Spoilage of Minced Pork and pH Data

The mean (± standard deviation, n = 4) initial TVC was 3.46 (± 0.34) log CFU/g (Figure 1). This is in accordance with data provided in the literature, i.e., total mesophiles ranging from 2.0 to 4.2 log CFU/g [24], as well as with EU legislative requirements; according the EC regulation 2073/2005 on microbiological criteria, the mean log aerobic colony counts of pork carcasses after dressing and before chilling should not exceed 4–5 log CFU/cm^2^) [25]. The microbial groups identified to mainly contribute to the spoilage of minced pork under the conditions of this study were *Pseudomonas* spp., *Br. thermosphacta* and LAB. Indeed, as illustrated in Figure 1, the microbiota of minced pork consisted of *Pseudomonas* spp., which were the dominant spoilage microorganisms, followed by *Br. thermosphacta* and LAB. Bacteria of the Enterobacteriaceae family, despite their relatively low initial level (i.e., mean (± standard deviation) initial population of 1.89 (± 0.45) log CFU/g), appeared to grow faster than the aforementioned microbial groups at higher temperatures (8 and 12 °C) and under dynamic conditions, reaching final population levels varying from 7.01 to 7.60 log CFU/g (Figure 1). *Pseudomonas* spp., *Br. thermosphacta*, LAB and Enterobacteriaceae have certainly been found to be associated with meat spoilage under various packaging and storage conditions [26,27]. On the other hand, yeasts demonstrated a relatively negligible contribution to minced pork spoilage (Figure 1), with the maximum population increase observed during storage at 8 °C and being 3.22 log CFU/g. Overall, the microbial evolution recorded during storage of minced pork under dynamic temperature conditions was comparable to that observed during storage at 8 °C (Figure 1).

Aiming at the more detailed quantitative description of the growth behavior of the dominant spoilage microorganisms of minced pork, the growth kinetic parameters for the abovementioned selected bacterial groups (i.e., *Pseudomonas* spp., *Br. thermosphacta*, LAB) as well as for the TVC were estimated and are presented in Appendix A. The mean estimated λ (h) for total mesophiles was 33, 18 and 16 at 4, 8, and 12 °C, respectively, and the corresponding mean μ_max_ values (h^−1^) were 0.079, 0.127, and 0.222.

Concerning the pH of the minced pork samples, the initial value (mean ± standard deviation, n = 4) of fresh samples was 5.59 ± 0.06, which is in agreement with the pH values reported for pork meat in the scientific literature [28]. The pH values recorded at the end of minced pork storage were 6.32 ± 0.39, 6.23 ± 0.44, 5.78 ± 0.42, and 6.08 ± 0.06 for storage at 4 °C, 8 °C, 12 °C, and dynamic temperature conditions, respectively (Figure 2). The pH changes in the minced pork samples during storage are expected to reflect the relative abundance and the metabolic activity of the dominant spoilage microorganisms; species belonging to the genus *Pseudomonas* are known for their proteolytic activity, resulting in the production of alkaline bacterial metabolites [29], whereas both *Br. thermosphacta* and LAB are well-established organic acids (acetic and/or lactic acid) producers and thus, contribute to a gradual pH decrease when allowed to grow at high populations [30,31].

### 3.2. Estimation of Minced Pork Spoilage Using Spectral Data

Typical FTIR, VIS, and MSI spectra corresponding to fresh and spoiled meat samples are shown in Figure 3a–c respectively, where the black solid line corresponds to fresh (3.0 log CFU/g), the dashed line to spoiled (10.3 log CFU/g) samples, while circles represent the selected x-variables as described earlier during the model development process. The feature selection scheme used for model development herein, accounts for the Martens algorithm, which is provided by the applied chemometric software and also used in Papadopoulou et al. [15]. Concatenated spectra used for data fusion are visualized in Figure 3d.

The FTIR spectrum in the range of 1800–900 cm^−1^ gives an overall fingerprint of the biochemical composition of food samples, also including the metabolic activity of microorganisms, information that is used by the models for the prediction of spoilage. This region has also been used for model development by other researchers [32,33,34,35]. The information provided by FTIR spectra, and their specific properties in terms of the selected wavenumbers, are in accordance to Papadopoulou et al. [15], who used the same food matrix and feature selection scheme. Some of the selected wavenumbers, which seem to contribute to spoilage are: (a) The major peak at 1650 cm^−1^ [36], corresponding to water-containing solid samples and primary amines; (b) certain features in the region 1314–1205 cm^−1^ ascribed to amide III (30% C-N stretch, 30% N-H bend, 10% C=O-N, 20% other) [35]; (c) 1240 cm^−1^ ascribed to C-N stretching from amines from amino acids [33]; (d) 1401 cm^−1^ to C=O symmetric stretch of COO- groups [35]; (e) 1413, 1405 cm^−1^ C-N from amides [34]; and (f) the region 1600–1700 cm^−1^ ascribed to amide I (80% C=O stretch, 10% C-N stretch, 10% N-H bend) [35]. Most of the above-mentioned features correspond to amides and amines and could be attributed to the indigenous proteolytic meat enzymes (autolysis) and to the microbial proteolytic activity occurring during storage [37].

With reference to the spoilage prediction models development for each sensor, first the FTIR data were used. The performance metrics of the root mean square error (RMSE, log CFU/g) and the coefficient of determination (R^2^) at the model training phase, suggest good prediction performance, with their values being 1.028 and 0.793, respectively (Table 1). Such good performance was further substantiated upon model validation using the external test data set; values of RMSE (log CFU/g) and R^2^ were 0.915 and 0.834, respectively. In addition, the slope parameter of the regression line, correlating the predictions and the observations at the validation phase (test set), had a value of 0.951, indicating an almost perfect one-to-one relationship between the predicted and measured (actual) TVC values (Figure 4). Similar workflow using PLSR modelling has been previously followed by Papadopoulou et al. [15] for FTIR spectral data of pork, according to their findings, a rather good prediction performance was attained on the test dataset, with a standard error of prediction 0.674 and correlation coefficient 0.880.

The same procedure, i.e., training, cross-validation, and external validation (testing) were applied to the acquired VIS data (Figure 5), and the corresponding models exhibited similar performance to that observed for the FTIR sensor, with RMSE (log CFU/g) and R^2^ values of 1.009 and 0.800, respectively (Table 1). The prediction on the test samples was slightly inferior to that of the FTIR model, i.e., with R^2^ = 0.788 and RMSE = 1.034 log CFU/g, albeit still acceptable as also supported by the slope parameter value of 0.843. At this point it should be noted that spectral data in the VIS region have not been used as extensively as FTIR and MSI data for the estimation of food spoilage, with their main applications including monitoring of salmon spoilage [38] and characterization of beef quality attributes (i.e., tenderness, pH, and colour) [39], as complimentary part of NIR spectra. Chen et al. [40] utilized the region from 530 to 920 nm to evaluate chicken carcasses freshness, while Grau et al. [41] used the region 400–1000 nm for freshness assessment in packaged chicken breasts. Furthermore, VIS spectra in the range of 340–850 nm have been used previously for shelf-life prediction of leafy vegetables [11].

In the case of the MSI data (Figure 6), the developed model did not perform as satisfactorily as the models based on the aforementioned sensors, with RMSE (log CFU/g) and R^2^ values of 1.173 and 0.727, respectively. Nevertheless, the latter performance indices’ values do not overrule the potential of MSI as a reliable source of information for minced pork microbiological spoilage. Indeed, MSI data have been shown to be promising with regard to monitoring of the microbiological quality of meat, using different data analysis approaches. As reported by Dissing et al. [42] MSI data were shown to be competent in providing estimates of the microbiological quality of pork, with the pertinent developed model predicting total viable counts with a standard error of prediction of 7.47%. MSI data of beef fillets were used for quality assessment by Tsakanikas et al. [43], and according to the calculated regression results (R^2^ = 0.98) good performance was attained. Similarly promising results with regard to the use of MSI in the estimation of beef fillets microbiological quality were reported by Panagou et al. [13]. Hence, as demonstrated by the findings of the present work and the aforementioned studies, MSI data constitute rather valuable means for the accurate estimation of meat quality.

Both VIS and MSI sensors contain the region of visible 400–850 nm, with the difference that MSI data provide not only chemical but also spatial information. In the region of visible, there are peaks (located in the 470–610 nm region) which are associated with myoglobin oxidation occurring during storage [44,45]. MSI spectra also cover the NIR region of 850–970 nm, that is associated with protein and moisture content [46,47]. Taking this into account, and also the fact that microbial metabolic activity is mainly associated with the region of infrared (as also commented above for FTIR) and relatively less with the visible region, an explanation could be provided regarding the better performance of the model based on the FTIR data as compared to the models based on the VIS or MSI data.

Finally, when “naïve” fusion of spectral data (Figure 7), as described earlier, was applied, the derived model appeared to exhibit an improved performance not only at the training phase, but also upon external validation (R^2^ = 0.862 and RMSE = 0.836 log CFU/g), except, for the FTIR case. Although not providing a great performance optimization advantage, the performed data fusion holds promise if other fusion schemes will be adopted, e.g., fusion on decision by each sensor and fusion of important features by each sensor. Data fusion of spectroscopy in tandem with chemometrics has been studied for several food-related applications [48,49], leading to the improvement of predictions. Herein, the applied preliminary data concatenation (as a fusion scheme) did not show significant improvement. Arguing about it, we can mention that, apart from the naïve nature of the fusion scheme adopted here, aiming at just showcasing its application, the predictive power of the individual sensors is quite high, especially in the case of FTIR, so the improvement is expected to be less significant, i.e., there is not enough room for major benefit. In general, fusion approaches are used in situations where individual/heterogeneous data do not result to such efficient models as the ones presented in this study.

## 4. Conclusions

Individual sensors and concatenated spectra showed good performance indices with RMSE (log CFU/g) of prediction of the validation dataset varying from 0.836 to 1.173 and R^2^ from 0.727 to 0.862. The data derived from FTIR and the fused data had better performances than the models derived from the VIS and MSI data, showing that the predictive power of the two latter sensors are inferior to the FTIR. Nevertheless, the predictive information provided by VIS and MSI sensors is adequate for applications in spoilage monitoring and within the accepted limits, as the latter are delineated by the classical microbiology limitations in TVC estimation. Overall, the spectroscopy-based sensors evaluated in the present study seem to give promising results with regard to the estimation of the microbial quality of minced pork. Spectroscopic methods, such as the ones evaluated in the present study, could be used as additional (to the microbiological analyses) information allowing for the rapid large-scale screening of numerous samples. Monitoring of production lines, or any other point of the food supply chain, via the utilization of low-cost, time-efficient, and non-invasive methods is anticipated to lead to the improvement of the end products, as it should be easier to apply timely corrective actions. In this sense, additional expected outcomes include reduction of economic losses and minimization of consumer complaints.

## Figures and Tables

**Figure 1 foods-08-00238-f001:**
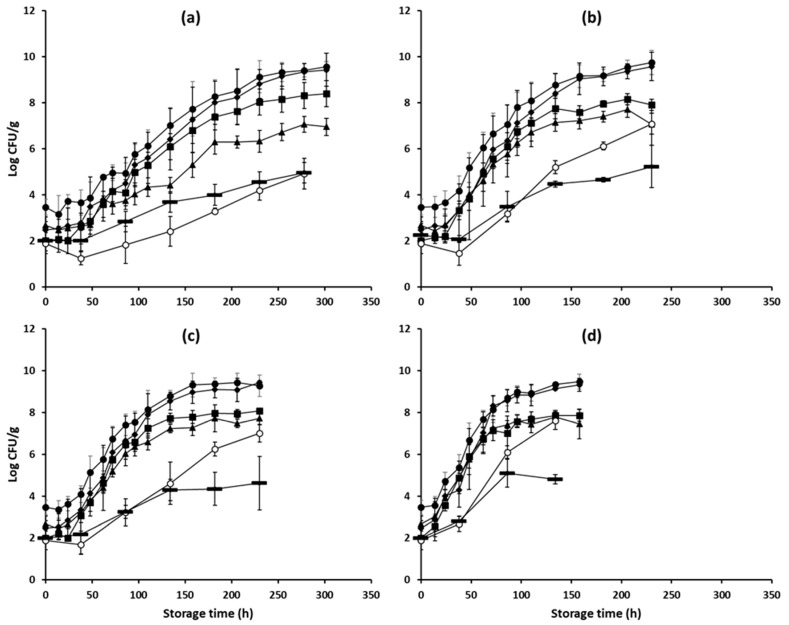
Mean (± standard deviation, n = 4) populations of different microbial groups in minced pork during aerobic storage at 4 °C (**a**), 8 °C (**b**), 12 °C (**c**), and dynamic temperature conditions (**d**). Total mesophiles (●), *Pseudomonas* spp. (♦), *Brochothrix thermosphacta* (■), Lactic acid bacteria (▲), Enterobacteriaceae (○) and Yeasts (▬).

**Figure 2 foods-08-00238-f002:**
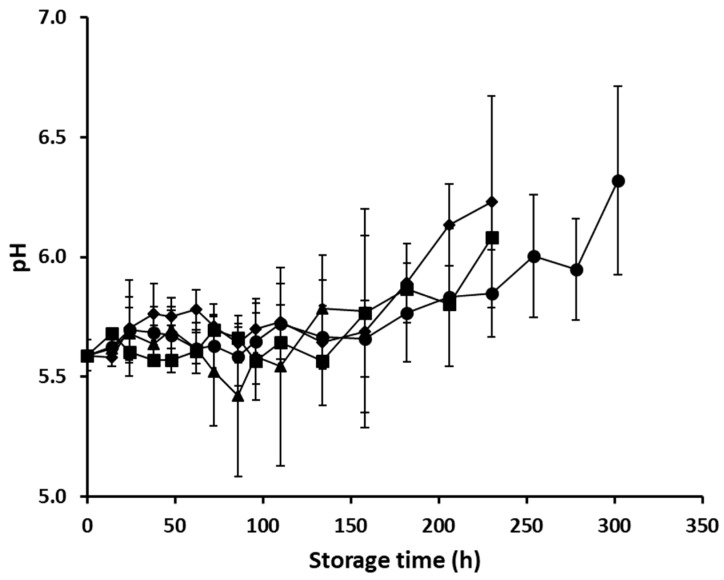
Values of pH (mean ± standard deviation, n = 4) of minced pork during aerobic storage at 4 °C (●), 8 °C (♦), 12 °C (▲), and dynamic temperature conditions (■).

**Figure 3 foods-08-00238-f003:**
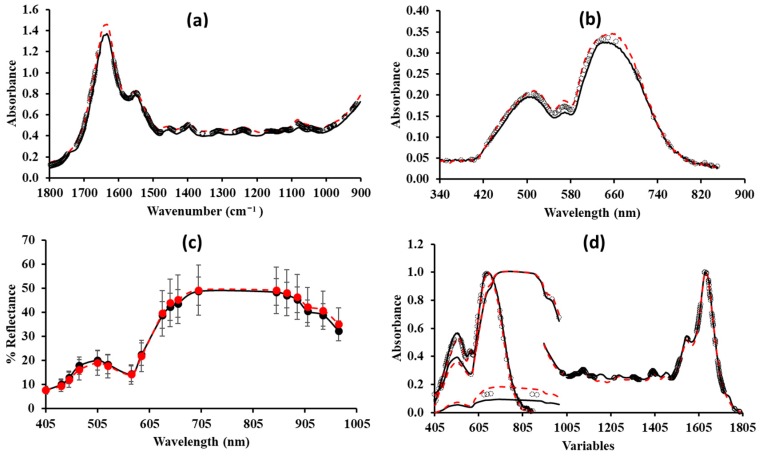
Representative Fourier transform infrared (FTIR) (**a**), visible (VIS) (**b**), multispectral imaging (MSI) (**c**) spectra, and data fusion spectra (**d**) corresponding to minced pork samples with low (fresh meat (3.0 log CFU/g), black solid line) and high (meat stored at 8 °C for 230 h (10.3 log CFU/g), red dashed line) microbial populations. In the case of FTIR, VIS and data fusion spectra, open symbols (o) correspond to selected features.

**Figure 4 foods-08-00238-f004:**
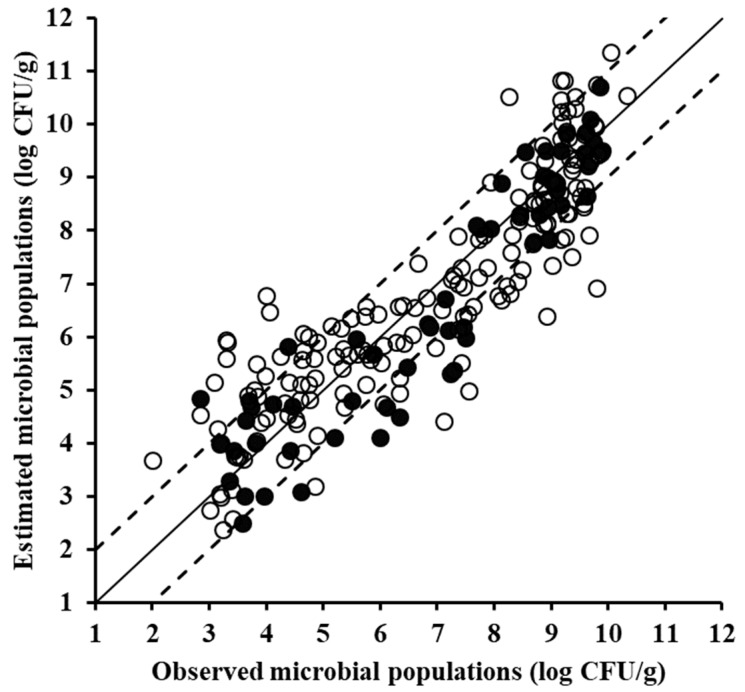
Comparison between the observed and estimated by the partial least squares regression (PLSR) model total mesophilic microbial populations based on Fourier transform infrared spectral data of minced pork samples for the training (solid symbols, 170 samples) and the validation (open symbols, 58 samples) datasets (solid line: the ideal *y* = *x* line; dashed lines: the ±1 log unit area).

**Figure 5 foods-08-00238-f005:**
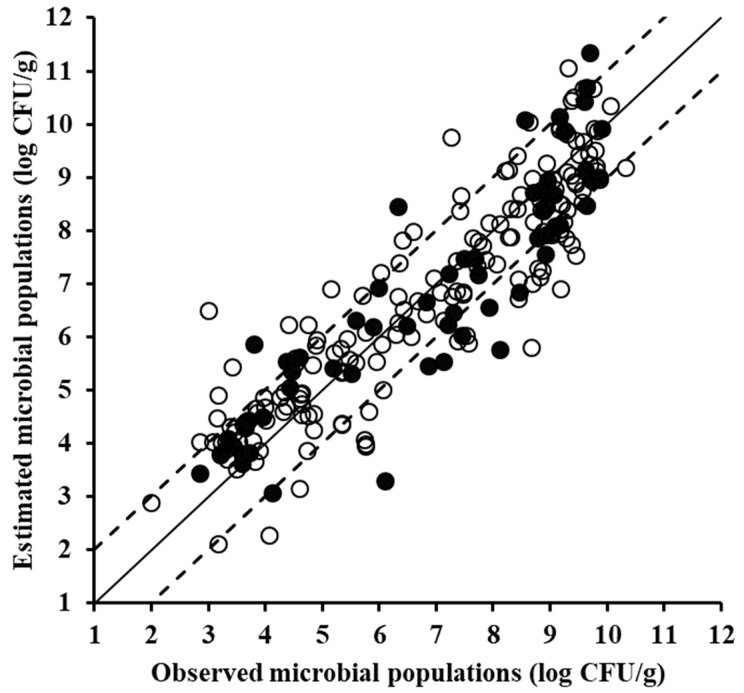
Comparison between the observed and estimated by the PLSR model total mesophilic microbial populations based on visible spectral data of minced pork samples for the training (solid symbols, 170 samples) and the validation (open symbols, 58 samples) datasets (solid line: the ideal *y* = *x* line; dashed lines: the ±1 log unit area).

**Figure 6 foods-08-00238-f006:**
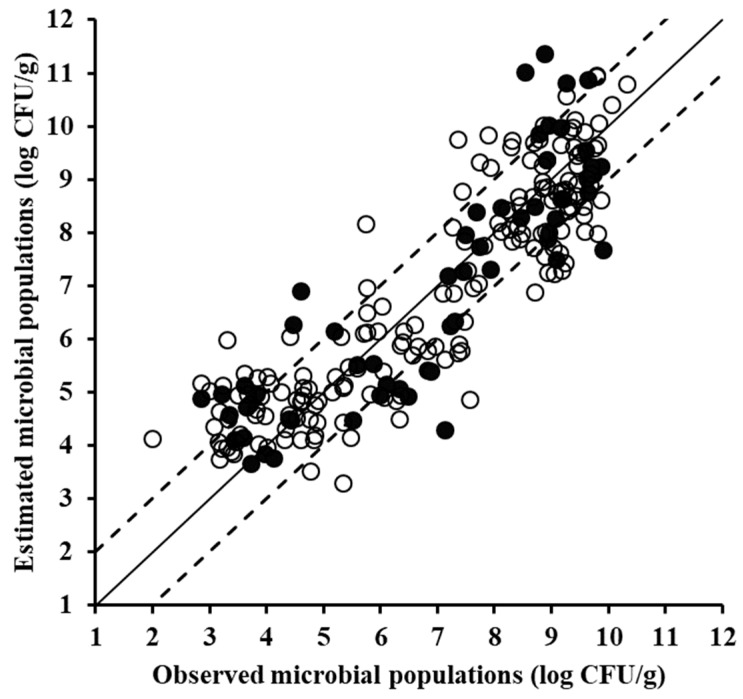
Comparison between the observed and estimated by the PLSR model total mesophilic microbial populations based on multispectral imaging spectral data of minced pork samples for the training (solid symbols, 170 samples) and the validation (open symbols, 58 samples) data sets (solid line: the ideal *y* = *x* line; dashed lines: the ±1 log unit area).

**Figure 7 foods-08-00238-f007:**
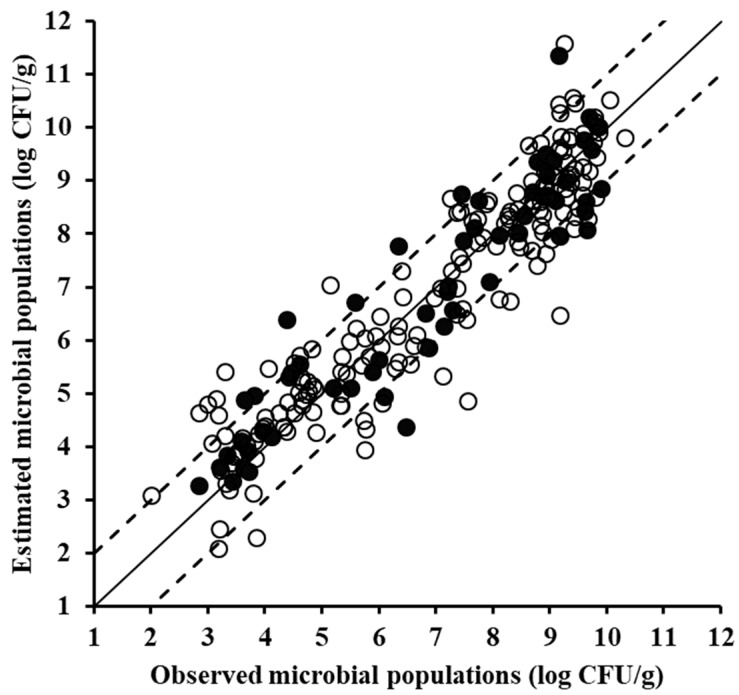
Comparison between the observed and estimated by the PLSR model total mesophilic microbial populations based on data fusion spectral data of minced pork samples for the training (solid symbols, 170 samples) and the validation (open symbols, 58 samples) data sets (solid line: the ideal y = x line; dashed lines: the ±1 log unit area).

**Table 1 foods-08-00238-t001:** Performance metrics of the PLSR models correlating total mesophilic microbial populations in pork samples on the basis of spectroscopic data, using distinct datasets for model training (n = 170) and validation (n = 58).

Type of Data	Data Set	Slope	Offset	R^2^	RMSE (Log CFU/g)
FTIR data	Training (n = 170)	0.844	1.068	0.844	0.886
	Cross-validation *	0.826	1.191	0.793	1.028
	Testing (n = 58)	0.951	0.099	0.834	0.915
VIS data	Training (n = 170)	0.860	0.960	0.860	0.840
	Cross-validation *	0.839	1.100	0.800	1.009
	Testing (n = 58)	0.843	0.992	0.788	1.034
MSI data	Training (n = 170)	0.841	1.092	0.841	0.896
	Cross-validation *	0.817	1.261	0.794	1.025
	Testing (n = 58)	0.834	1.201	0.727	1.173
Data fusion	Training (n = 170)	0.909	0.622	0.909	0.676
	Cross-validation *	0.893	0.720	0.864	0.834
	Testing (n = 58)	0.883	0.861	0.862	0.836

* Leave-one-out cross-validation. FTIR: Fourier transform infrared; VIS: visible; MSI: multispectral imaging; R^2^: coefficient of determination; RMSE: root mean square error (log CFU/g).

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
