# Peer review of "Estimation of Minced Pork Microbiological Spoilage through Fourier Transform Infrared and Visible Spectroscopy and Multispectral Vision Technology"

_foods, 2019, doi:10.3390/foods8070238_

Reviewer 1 Report

This paper covers a topic of interest. Although numerous studies exist on the effect of different storage temperature on microbiology growth and spoilage, the estimation of microbiological quality of pork using non-invasive spectroscopy-based sensors are under explored. Therefore, I do think the paper presents new findings. However the paper needs some work. Please see the comments below.

Line 68-69 Two independed minced pork batches - in the opinion of the reviwer it is not enough repetition unless they are preliminary tests.

Line 70-72 "depending on the applied storage temperature" - please explain

Line 61 Please give the pate production process.

Line 181 The author should explain this clearly since not all countries have this regulation. Please add reference.

Line 295 Since the major aim of study is to estimation of the microbial population of minced pork using FTIR, VIS, MSI the author needs to further discuss the potential behind mechanism about the changes of parameter instead.

Line 337 Please add practical aplication of this research in your conclusion.

Author Response

Line 68-69 Two independed minced pork batches - in the opinion of the reviwer it is not enough repetition unless they are preliminary tests.

Although preliminary tests were not conducted for present study, similar storage experiments for pork (and other food products) have been conducted by our research group, providing data substantiating the applied experimental design. Parameters such storage conditions (i.e. temperature), sample size, biological variability and experimental uncertainty were all taken into account in the experimental design, while time efficiency and resulting measurements’ accuracy and credibility were also tried to be assured to the greatest possible extent. Specifically, given the objectives of the present study and the data generated in previous studies, two independent experiments (in terms of time and minced pork batch) and two distinct samples (i.e. minced pork patties) being analysed at each applied storage condition and sampling interval, as applied herein (sample size of n=4), were regarded as sufficient (under the conditions of this study) to address the stated objectives and assess biological/experimental variability.  In total 228 samples were subjected to microbiological analyses, which are laborious and time-consuming, and to parallel measurements with three different sensors (i.e. FTIR, VIS and MSI). These 228 samples came from two independent experimental replicates (distinct time and minced pork batches), in order to cover biological variability, and within each replicate two distinct minced pork samples were subjected to the aforementioned analyses at each sampling interval. Also, different temperature conditions were used, covering storage of meat at refrigeration temperatures (4°C), thermal abuse (12°C), as well as intermediate (8°C) and more “realistic” temperature conditions (dynamic storage temperature). Nonetheless, the authors acknowledge the value of additional experimental replicates and more extensive datasets for the substantiation and further improvement of the developed models, which is something that future research will certainly address in order for the studied spectroscopic analytical methods to be efficiently integrated in the actual processing pipelines of food manufacturing environments.

 Line 70-72 "depending on the applied storage temperature" - please explain

The reviewer’s comment was taken into account and the corresponding part has been revised for clarification purposes to the following: “…specifically, analyses were carried out at 10- or 14-h intervals until 110 h of storage (sampling times: 0, 14, 24, 38, 48, 62, 72, 86, 96 and 110 h), and at 24-h interval thereafter. The experiments were concluded (i.e. final time points of analyses) at 158, 230, 302 and 230 h of storage at 4 °C, 8 °C, 12 °C and dynamic temperature conditions, respectively.”

 Line 61 Please give the pate production process.

More details regarding the process have been provided in the revised manuscript: “Minced pork was stored at 0 °C until the packaging process was completed. Packaging took place under a laminar flow cabinet, in order for aseptic conditions to be ensured. Meat portions of ca. 100g were weighted, patties were prepared…”.

 Line 181 The author should explain this clearly since not all countries have this regulation. Please add reference.

This refers to EU legislative requirements (and not to a national-level one), and more specifically to the EC No 2073/2005 regulation regarding microbiological criteria. In order for this to be clear, a pertinent revision has been made and the corresponding reference has been included in the revised manuscript.

 Line 295 Since the major aim of study is to estimation of the microbial population of minced pork using FTIR, VIS, MSI the author needs to further discuss the potential behind mechanism about the changes of parameter instead.

In order to address the reviewer’s comment, the following part has been included in the discussion of the revised manuscript: “Both VIS and MSI sensors contain the region of visible 400-850nm, with the difference that MSI data provide not only chemical but also spatial information. In the region of visible, there are peaks (located in the 470-610 nm region) which are associated with myoglobin oxidation occurring during storage [44, 45]. MSI spectra also cover the NIR region of 850-970 nm, that is associated with protein and moisture content [46, 47]. Taking this into account, and also the fact that microbial metabolic activity is mainly associated with the region of infrared (as also commented above for FTIR) and relatively less with the visible region, an explanation could be provided regarding the better performance of the model based on the FTIR data as compared to the models based on the VIS or MSI data.”

 Line 337 Please add practical aplication of this research in your conclusion.

Taking into account the reviewer’s comment, the following sentences have been included in the revised manuscript: “Spectroscopic methods, such as the ones evaluated in the present study, could be used as additional (to the microbiological analyses) information allowing for the rapid large-scale screening of numerous samples. Monitoring of production lines, or any other point of the food supply chain, via the utilization of low-cost, time-efficient and non-invasive methods is anticipated to lead to the improvement of the end products, as it should be easier to apply timely corrective actions. In this sense, additional expected outcomes include reduction of economic losses and minimization of consumer complaints.”

 Reviewer 2 Report

In the article presented for review, the possibilities of using modern non-invasive methods to detect changes leading to the deterioration of meat raw material were discussed - a very interesting topic that aroused the interest of the meat industry.
The manuscript is prepared very carefully and does not require a significant correction. The applied research methods as well as data processing do not raise any objections. Minor comments and suggestions concern refinement.

Author Response

Responses to the comments raised by Reviewer #2 have been provided in the attached pdf file of the manuscript (as replies to each one of the in-text comments).
